# Optimization of Artificial Neural Network (ANN) for Maximum Flood Inundation Forecasts

**Hongfei Zhu** [1,*] , **Jorge Leandro** [2] **and Qing Lin** [1]

1   Chair of Hydrology and River Basin Management, Department of Civil, Geo and Environmental Engineering, Technical University of Munich, Arcisstrasse 21, 80333 Munich, Germany; tsching.lin@tum.de
2   Chair of Hydromechanics and Hydraulic Engineering, Faculty IV School of Science and Technology, University of Siegen, Paul-Bonatz-Str. 9-11, 57068 Siegen, Germany; Jorge.Leandro@uni-siegen.de
*   Correspondence: Hongfei.Zhu@tum.de; Tel.: +86-131-1811-1818

**Abstract:** Flooding is the world's most catastrophic natural event in terms of losses. The ability to forecast flood events is crucial for controlling the risk of flooding to society and the environment. Artificial neural networks (ANN) have been adopted in recent studies to provide fast flood inundation forecasts. In this paper, an existing ANN trained based on synthetic events was optimized in two directions: extending the training dataset with the use of hybrid dataset, and selection of the best training function based on six possible functions, namely conjugate gradient backpropagation with Fletcher–Reeves updates (CGF) with Polak–Ribiére updates (CGP) and Powell–Beale restarts (CGB), one-step secant back-propagation (OSS), resilient backpropagation (RP), and scaled conjugate gradient backpropagation (SCG). Four real flood events were used to validate the performance of the improved ANN over the existing one. The new training dataset reduced the model's rooted mean square error (*RMSE*) by 10% for the testing dataset and 16% for the real events. The selection of the resilient backpropagation algorithm contributed to 15% lower *RMSE* for the testing dataset and up to 35% for the real events when compared with the other five training functions.

**Keywords:** ANN; flood forecast; maximum inundation area; optimization; training function





## 1. Introduction

More than 50 percent of the fatalities and 30 percent of the economic losses due to natural disasters are related to flooding [1]. Initial estimates put the damage of the floods in mid-July 2021 in North Rhine-Westphalia alone at more than 13 billion euros. Flooding from extreme rainfall events can occur anywhere [2]. As climate change is taking place, floods tend to cause even more serious consequences [3]. The recent IPCC report [4] emphasised once again the role of human activity in climate change. Flood forecasting is an effective, non-structural flood protection measure that can reduce both economic and life losses. Providing a 2D visual representation of flooding with enough lead time can make authorities better prepared for this natural disaster.

There are various numerical models available for urban flood protection [5]. Hydrological rainfall-runoff models can be generally classified into one-dimensional (1D) models, two-dimensional (2D), and 1D-2D coupled models. The two latter types of models are particularly suitable for urban flood inundation predictions [6]. However, they are also computationally demanding, which often restricts their application in real-time forecasting [3]. High-performance computing has made a big leap in recent years, bringing the possibility of faster 2D simulation for larger areas [7]. However, real-time 2D flood early warning systems (EWS) are still a challenge [6].

Data-driven models can be a good substitute for physically based flood models [8]. Data-driven models are founded on the analysis of the data about the target system, finding connections between the system state variables without explicit knowledge of the physical behaviors of the system [9]. Data-driven models only require the input and output data and

do not make use of physically based parameters, which alleviates the burden on the users for data gathering and model setup [6]. In terms of model performance, data-driven models exhibit high performance even for non-linear problems [10]. However, data-driven models are prone to over-fitting or under-fitting, inadequate training data, or bias [11]. ANN is a popular method in flood forecasting [12,13]. Dawson et al. (2001) [14] applied ANN to conventional hydrological models in a flood-prone study area in the UK. Since then, studies on flood forecasts with data-driven models started to thrive [15,16]. Thirumalaiah [17] compared the forecast of water levels along a river using backpropagation (BP), conjugate gradient, and cascade correlation ANN models. Bustami et al. (2007) [18] applied BP ANN model in order to predict the water level at gauge stations. Taghi et al. (2012) [19] employed the BP ANN and a time lag recurrent network to predict the water level at gauge stations and reached a similar performance. Humphrey et al. (2016) [20] combined a Bayesian ANN with a conceptual rainfall-runoff model to improve the accuracy of the neural network. Kasiviswanathan et al. (2016) [21] adopted ANN to forecast streamflow for flood management. Bermúdez (2019) [22] presented a rapid flood inundation forecast model using support vector machine (SVM) for hazard mapping. Sit and Demir (2019) [23] enhanced the forecasting results by adding more river network location information in which they used a discretized neural networks for the entire river network. Chang et al. (2018) [15] applied ANN to produce flood inundation maps in real time. Chu et al. (2020) [24] brought up an ANN-based emulation model framework for flood inundation modeling in order to reduce the simulation time. Tamiru et al. (2021) [25] presented a new way of combining data-driven and hydraulics model by using ANN to generate the runoff time series, which is then used as input for HEC-RAS 2D model to map the flood inundation. Zhou et al. (2021) [26] developed a framework for rapid flood inundation modelling that is comprised of a spatial reduction and reconstruction (SRR) module and a deep learning module. Lin et al. (2020) [6] proposed an ANN model to produce flood inundation maps in urbanized area with high-resolution from multiple inflow data.

One of the most influential factors of ANN performance is the training dataset. Ideally, a real-events-based dataset should be used if enough measurements are available. In the ANN model for flood forecast proposed by Kimura et al. (2020) [27], the hourly data for rainfalls and water levels at two different locations from 1992 to 2017 and from 2000 to 2019 were used. Kao et al. (2020) [28] produced multi-step-ahead flood forecasts with a dataset collected from 23 typhoon events with hourly hydrological data. Historical observation of inundation depths in urban areas is rare [29]. Therefore, a physical-based hydraulics model can be the best substitute to generate the synthetic hydrographs of flood depths for various storm events [15]. In the ANN model proposed by Chang et al. (2018) [15], InfoWorks ICM was used to create the dataset to train the ANN model. In the flood EWS developed in Bhola et al. (2018) [30], HEC-RAS 2D was used to create a database consisting of synthetic inundation maps. Crotti et al. (2020) [31] improved both the accuracy and reliability of the flood EWS in Bhola et al. (2018) [30] by adding a real-events-based dataset to the original synthetic dataset.

Another important variable of ANN architecture is the training function. A wide variety of training functions for ANN are available. Each function has its strengths and weaknesses. However, there is no single training function that works the best on all problems [32]. Hence, the evaluation of different training functions on a specific problem is necessary. Noori R. et al. (2010) [33] investigated the prediction of river flow with several functions; it turned out that the conjugate gradient backpropagation with Fletcher–Reeves updates (CGF) algorithm, scaled conjugate gradient backpropagation (SCG) algorithm, and the one-step secant backpropagation (OSS) algorithm had close performance and outperformed the other training functions involved in the comparison for multivariate linear regression models. In the ANN model for the classification of insulators in electrical power systems by Stefenon et al. (2020) [34], the conjugate gradient backpropagation with Powell–Beale restarts (CGB) algorithm was faster than the CGF algorithm, and the conjugate gradient backpropagation with Polak–Ribiére updates (CGP) algorithm was more

accurate than CGF algorithm. Lin et al. (2020) [6] showed that resilient backpropagation (RP) was more accurate and more general than the CGF method on producing flood inundation forecasts.

In this paper, we focused on optimizing the performance of ANN by improving the training dataset and selecting the most suitable training function. Section 2 presents the methodology for the optimization of the ANN, Section 3 compares the performance of the model before and after optimization, Section 4 discusses the results, and Section 5 draws a conclusion for this study.

## 2. Method

### 2.1. ANN Model for Forecasting Maximum Urban Flood Inundation

There are in total 430,485 pixels covering the study area, which is separated into a 50 by 50 mesh in even rectangular size, resulting in 2500 grids in total. The grid size was selected in accordance to the performance analysis in Lin et al. (2020) [6]. All the pixels were assigned to a corresponding grid according to their geographical locations. In total, there are 580 grids inside the case study floodable domain. For each of the grids, one ANN was trained; the inputs were the hydrographs of the rivers that contribute to the flooding in the urban area, and the outputs were the water depths at each pixel in all grids (see Figure 1). After that, the appropriate model architecture was determined by varying the number of hidden layers from 2 to 12 and the neurons number in each layer from 10 to 60, as in Lin et al. [6]. Going beyond those ranges showed no improvement on the performance of the ANN. As a result, 580 independent ANN were trained with the same training function and input river hydrographs. The same hydrographs were used as input for all the ANN to ensure an identical ANN topology so as to avoid sudden inconsistency of the simulated water depths at the borders of the grids of the ANN [24]. This assures that the inundation map, which is the combination of outputs from different ANN, is consistent. A detailed description of the model can be found in Lin et al. (2020) [6]. The inputs, outputs, and training function for the ANN are further described and explained in Sections 2.3.1 and 2.3.2.

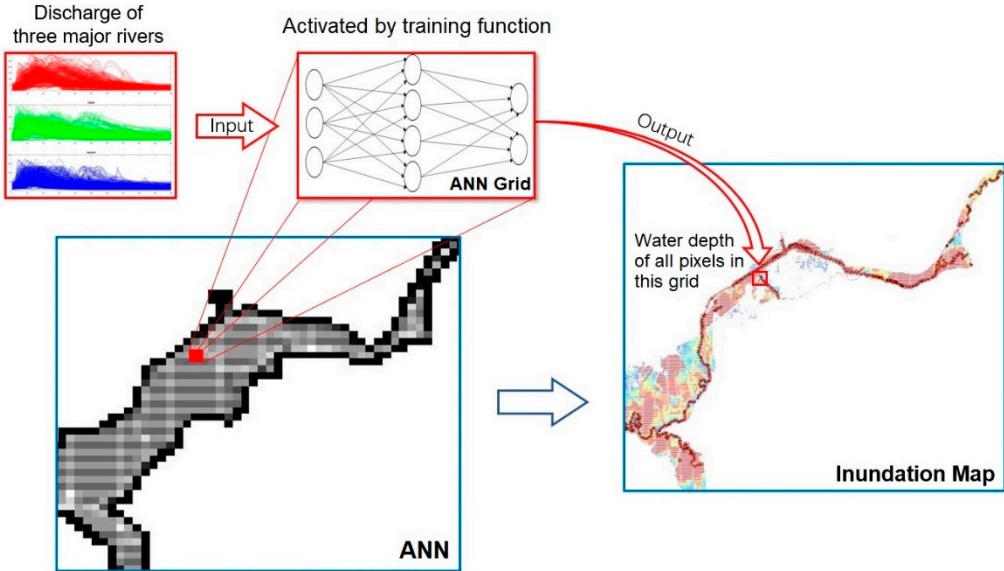

**Figure 1.** Structure and mechanism of each ANN for maximum flood inundation area forecasting.

### 2.2. Evaluation Methods

The dataset for the ANN model is divided into three parts: training, validating, and testing dataset. After the ANN was trained and validated with the first two datasets, it was tested on the testing dataset for further comparison. In this paper, the division ratio of

train: validation: test dataset is equal 4:1:1. *RMSE* was adopted to express the accuracy of the simulation. Computation of *RMSE* follows the function below:

$$RMSE = \sqrt{\frac{1}{m}\sum_{i=1}^{m}(T-S)^2} \tag{1}$$

where *T* is the measurements in the dataset, *S* is the simulated value, and m refers to the number of data pairs involved in a specific grid.

The *RMSE* of each grid in each event was calculated. The resulting maps' features are depicted in Figure 2. Furthermore, to assess the overall performance of the ANN on the entire testing dataset, the average *RMSE* map of all events was calculated, which is simply the average of all maps in Figure 2. The average *RMSE* in each grid of all testing events were computed with the following function:

$$avgRMSE(i,j) = \frac{1}{N}\sum_{n=1}^{N}RMSE_n(i,j) \tag{2}$$

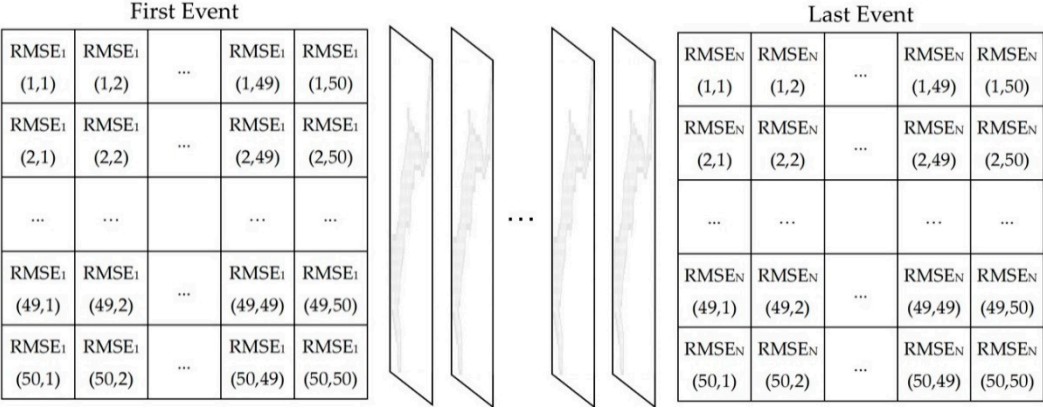

**Figure 2.** The *RMSE* maps for all the single events in the dataset. The *RMSEn (i, j)* is the *RMSE* value in the grid at *i*-th row and *j*-th column for the event number, *n*.

*N* denotes the number of all the events in the dataset. $RMSE_n(i,j)$ represents the *RMSE* value in the grid at *i*-th row and *j*-th column for event number, *n*.

Major results are presented in the form of differences between the average *RMSE* maps to assess the performance achieved via the optimization. Five statistical parameters are introduced. Sum of *RMSE* is the sum of *RMSE* in all the grids. Improved Grids is the number of grids that show a positive value in the difference of *RMSE* (yellow grids). In addition, in the parentheses, the number of grids that exceed the maximum value of the color bar is shown. The same strategy was also applied to Deteriorated Grids. Improvement is the sum of the values in all Improved Grids. Similarly, the same strategy is also applied to Deterioration. A parameter called Change Ratio was introduced to evaluate the overall change in performance; the function is defined by:

$$\text{Change Ratio} = \frac{\text{Improvement} - \text{Deterioration}}{\text{Sum of } RMSE} \tag{3}$$

*2.3. Optimization of the ANN*

2.3.1. Training Dataset

In this study, a total of 360 flood events were used. A total of 180 are synthetic flood events produced by the FloodEvac tool [30], and 180 are additional real flood events. All were generated by HEC-RAS 2D [35]. The synthetic database was created within two steps. Firstly, the hydrologic model LARSIM (Large Area Runoff Simulation Model) [36],

which is used for flood forecasting at the Bavarian Environment Agency [37], was used to calculate the hydrographs of the discharge of the rivers. After that, the 2D hydraulic model HEC-RAS 2D was used to simulate 180 advective and convective events, which were then used as the flood inundation maps of the database. For further details on the generation of the synthetic database, please refer to the work of Bhola et al. (2018) [30]. Crotti et al. (2020) [31] extended the FloodEvac database by adding 180 rescaled historical flood events, which contribute to a hybrid database containing 360 events. The 180 events in the real-event-based database were generated through normalization, rescaling, and generation based on the full discharge series for the gauges of the three major rivers from 1970 to 2017. The hydrographs of three rivers from both databases are shown in Figure 3. For further details on the datasets, please refer to the work of Crotti [31].

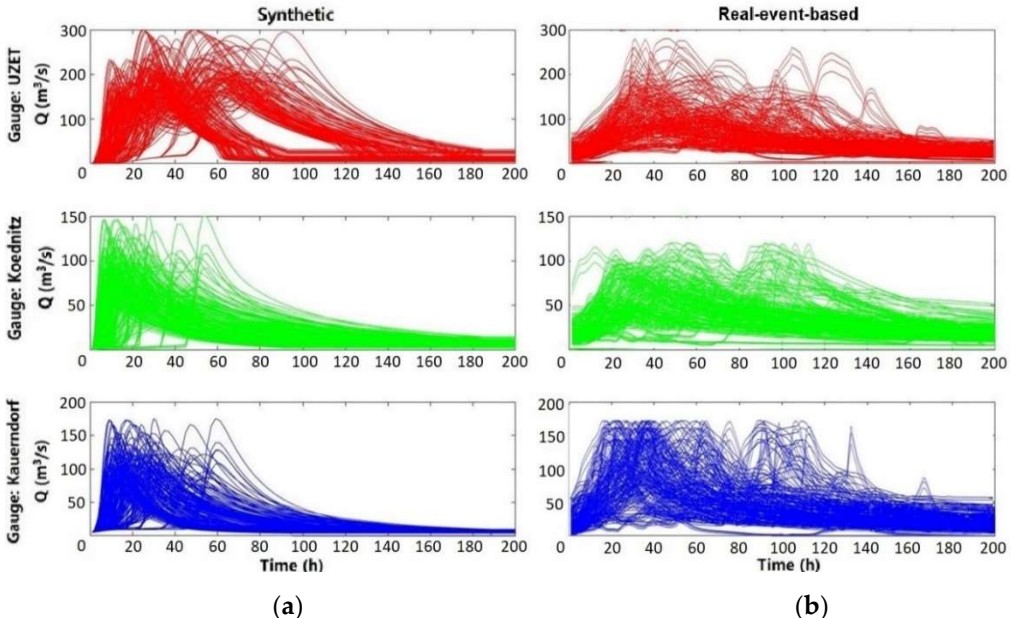

**Figure 3.** The hydrographs of three major rivers of (**a**) synthetic database and (**b**) real-event-based rescaled historical database.

The distribution pattern of the flow rate of the events in the real-event-based database and synthetic database are shown in Figure 4. It can be seen that they differ across all the gauges between the two databases. For the gauge Kauerndorf and Unternzettlitz, the peaks are close to the normal distribution for both the databases. For the gauge Ködnitz, the peak flow rate of the real-event-based database is on average higher than that of the synthetic database. This is mainly due to the fact that the real-event-based database takes the Ködnitz gauge as the most important one. This gauge monitors the river that crosses the study area of Kulmbach and has the higher discharge. The synthetic database does not take this feature into account.

Crotti et al. (2020) found that using the hybrid database leads to a more accurate prediction of the FloodEvac framework than using solely the synthetic database [31]. It was expected that the same effect could be achieved herein. Therefore, the performance of the ANN trained with the hybrid database was expected to improve.

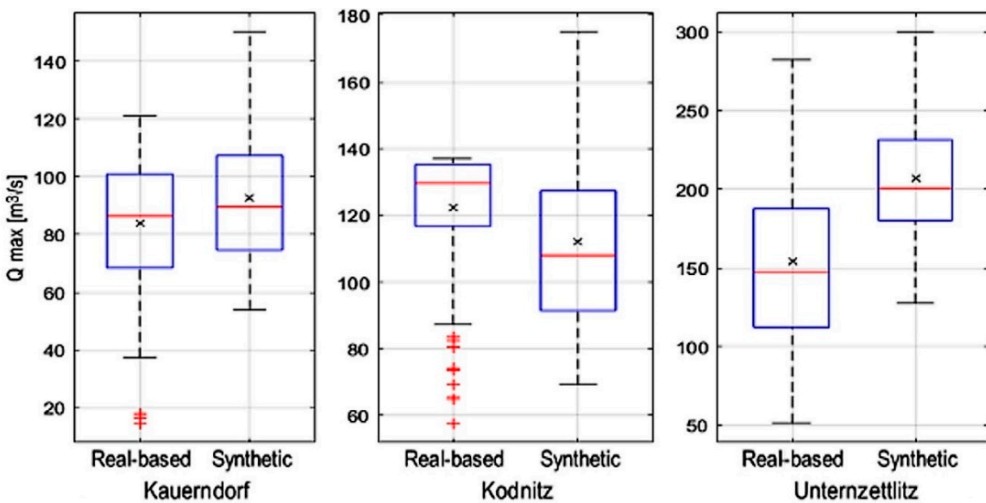

**Figure 4.** Comparison between the synthetic database and maximum flow rate of the gauges of the three major rivers for the events in the real-event-based database (Taken from [31]).

### 2.3.2. Training Function

Backpropagation is the most commonly used algorithm for supervised machine learning with ANN in a multi-layer structure. It was first presented by Rumelhart et al. (1985) [38] and further developed by Riedmiller and Braun (1993) [39]. To make the differences between the network outputs and target values as small as possible, the idea behind backpropagation is to use the rule of chain to modify the parameters from the output layer to the input layer. In this case, the weights concerning the loss function *L* are computed as:

$$\frac{\partial L}{\partial w_{ij}} = \frac{\partial L}{\partial O_i} \cdot \frac{\partial O_i}{\partial net_i} \cdot \frac{\partial net_i}{\partial w_{ij}} \tag{4}$$

where $w_{ij}$ is the weight from *i*-th neuron to *j*-th neuron, *L* is the loss function of the model, $O_i$ is the output of the model, and $net_i$ is the weighted sum of the inputs of neuron *i*.

The minimization of the loss function can be accomplished by the gradient descent algorithm if the partial derivative of the weight is known:

$$w_{ij}(t+1) = w_{ij}(t) - \epsilon \cdot \frac{\partial L}{\partial w_{ij}} \tag{5}$$

The learning rate $\epsilon$ has a great impact on the time of reaching convergence. When $\epsilon$ is too large, the optimum solution will be missed, while gradient descent will take too much time and space when $\epsilon$ is too small.

Many training functions can be used to combine the gradient descent principle in backpropagation. According to the "no free lunch" theorem by Wolpert and Macready, it is hard to decide which one is the best training function for a specific problem without experiment [32,40]. Especially for a complex process like flood inundation forecast in an urban area, as the quantity of data is large, the time series are of various types, and the urban land surface features are complex. Among all the popular training functions, some were excluded due to overwhelming computation storage demand under the current model setup, like the Levenberg–Marquardt algorithm; some were taken out because newer versions are available (e.g., gradient descent methods). The following algorithms were therefore chosen: conjugate gradient backpropagation with Fletcher–Reeves updates (CGF) [41], conjugate gradient backpropagation with Polak–Ribiére updates (CGP) [42], conjugate gradient backpropagation with Powell–Beale restarts (CGB) [43], one-step secant backpropagation (OSS) [44], resilient backpropagation (RP) [39], and scaled conjugate gradient backpropagation (SCG) [45]. CGB, CGP, and CGF belong to conjugated gradient

methods, while the other three functions are backpropagation methods. In this paper, six ANN were trained with different training functions. Their performances were compared with each other on both testing dataset and real events to find the most suitable for urban flood forecasting of the maximum flood inundation extent.

1.  Conjugate gradient backpropagation with Fletcher–Reeves updates (CGF)

The conjugated gradient method was firstly presented by Magnus R. et al. in 1952 [46]. After that, many conjugated gradient methods were developed upon it. Regardless of type of conjugate gradient method used, the search direction of the algorithm is reset periodically to the negative value of the gradient. The standard reset takes place when the number of iterations achieves the number of the parameters in the network. There are other reset methods available that bring different benefits to the training process. Therefore, various conjugate gradient methods are developed and can be distinguished by the reset method [46].

In the method with Fletcher–Reeves updates (1964), the update procedure was carried out using the ratio of the norm squared of the current gradient to the norm squared of the previous gradient [41]. The CGF is usually faster than RP on some problems. It requires more storage than a simpler algorithm like the gradient descent method. Therefore, this algorithm is suitable for networks with many weights [46].

2.  Conjugate gradient with Polak–Ribiére updates (CGP)

The reset method of Polak–Ribiére [42] is the inner product between the present gradient and the former gradient change divided by the norm squared former gradient. The CGP method generally has a performance similar to CGF. However, the storage requirements of the CGP method (four vectors) are somewhat higher than that of CGF method (three vectors) [42,47].

3.  Conjugate gradient with Powell–Beale restarts: CGB

The resets of the Powell–Beale restart method [43] occur if there is little orthogonality left between the previous and the current gradient. In that case, the search direction will be reset to the negative value of the current gradient. The CGB routine has better performance than CGP on certain problems. On the other hand, the storage required by CGB (six vectors) is larger than that for CGP (four vectors) [43].

4.  One-step secant backpropagation: OS

OSS method by Battiti et al. in 1992 tries to bridge the gap between the quasi-Newton algorithms and the conjugate gradient algorithms [44]. Unlike quasi-Newton algorithm, this algorithm does not store the complete Hessian matrix. Instead, it assumes that the Hessian matrix of the previous iteration is the identity matrix of the current iteration. This is beneficial because the new search direction can be determined without computing a matrix inverse. It requires more computation time and storage per epoch than the conjugate gradient algorithms to a small extent. For more information, please refer to [44].

5.  Resilient backpropagation (RP)

RP was developed by Riedmiller M. and H. Braun [39]. The target of the resilient backpropagation (RP) algorithm is to remove the drawback of the magnitudes of the partial derivatives. This means that only the sign of the derivatives of the performance function controls the update direction of the weight, and the magnitude of them does not matter. The weight-change size is decided by an adaptive, three-step procedure. In a first step, when the derivative regarding that weight shows the same sign in two consecutive iterations, the value for each weight and bias increases. In a second step, when the sign of the derivatives with regard to the weight changes from the previous iteration, the value for each weight and bias decreases. In a third step, when the derivative is zero, the value remains the same. Further detail can be found in [39].

6. Scaled conjugate gradient backpropagation: SCG

SCG was proposed by Moller et al. [45]. It is a fully automized algorithm; it includes non-user-dependent parameters and forgoes time-consuming line search. It can be used to train any network as long as the derivatives of its weight, net-input, and transfer functions exist. Backpropagation algorithm is used in this algorithm to compute the derivatives of the performance function concerning the weight and bias variables. Just like CGP, CGF, and CGB, the scaled conjugate gradient algorithm is based on conjugate directions except this algorithm does not carry out line search for each iteration. More information about this algorithm can be found in [45].

### 2.4. Study Area and Real Events

The study area is the city of Kulmbach, located in Bavaria, Germany. The river White Main crosses and divides the city into northern and southern sections. There are a total of 25,700 habitants in an area of 92.77 km$^2$ [6]. There are in total four real events available with measurements of both inundation area and discharge in three major rivers. These rivers were identified as the main contributors to flooding in Kulmbach in the work of [30]. These rivers are measured by gauges Unternzettlitz, Ködntiz, and Kauerndorf (Figure 5), respectively. These real flood events occurred in 2005, 2006, 2011, and 2013. Both the duration and the maximum flow rate of these real events in three major rivers are distinctly different from each other (Figure 6).

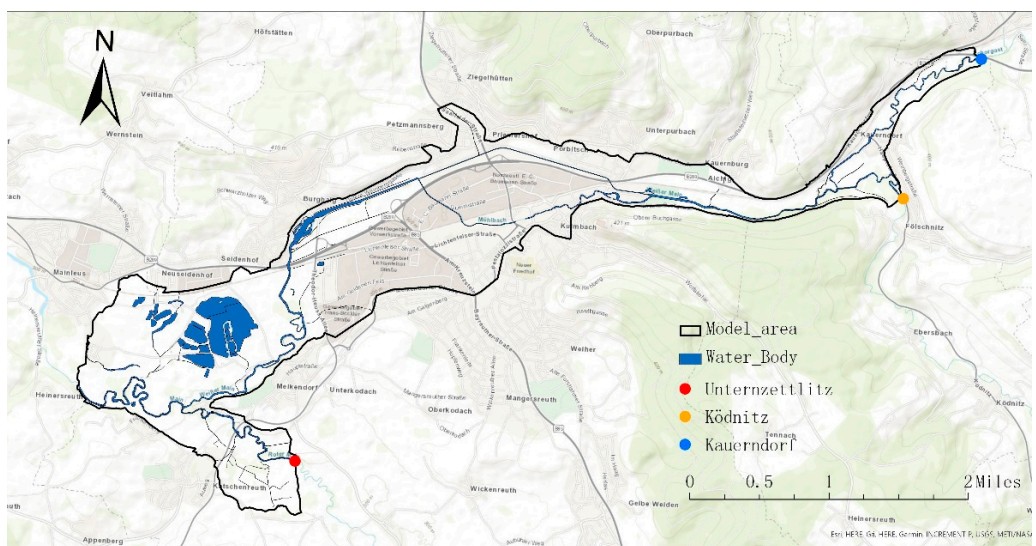

**Figure 5.** Gauges and rivers of the study area of Kulmbach.

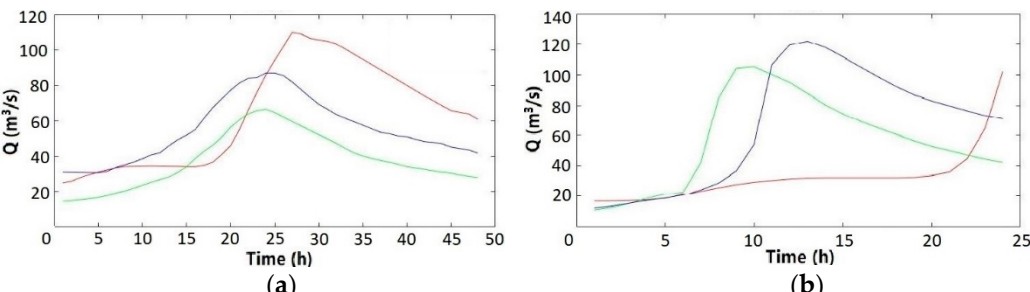

**Figure 6.** *Cont.*

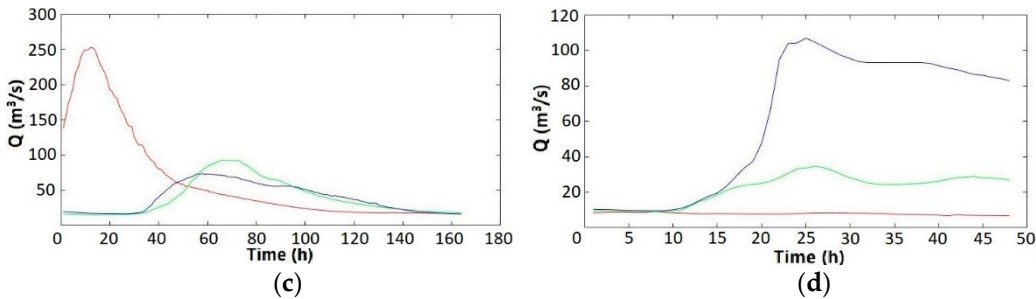

**Figure 6.** The hydrographs of three major rivers (red: Unternzettlitz, green: Ködntiz, blue: Kauerndorf) in four real events (**a**) 2005, (**b**) 2006, (**c**) 2011, and (**d**) 2013.

## 3. Results

### 3.1. Performance on the Testing Dataset

In total, six ANN were trained by different functions with a hybrid database. The performance was first quantified on the testing dataset and was presented in the form of average *RMSE* maps. The average *RMSE* maps of the synthetic-databased trained with RP was used as the benchmark [6,30]. Since the difference maps between the mean *RMSE* are visually difficult to distinguish, several parameters were introduced to present the comparison. All the parameters of the resulting maps are summarized in Table 1. Auxiliary maps are shown in the figures in the Appendix A Figure A1.

**Table 1.** Statistics of the *RMSE* difference map for ANN on the testing dataset. Only the benchmark ANN, RP (SD), was trained with the synthetic database. All other ANN were trained with the hybrid database.

| Set Name | Sum of *RMSE* (m) | Improved Grids [1] | Deteriorated Grids [2] | Improvement [3] (m) | Deterioration [4] (m) | Change Ratio [5] |
|---|---|---|---|---|---|---|
| RP (SD) | 50.08 | - | - | - | - | - |
| RP | 44.83 | 377 (21) | 110 (3) | 6.51 | 1.41 | +10.23% |
| CGB | 49.20 | 270 (14) | 217 (6) | 3.61 | 2.88 | +1.46% |
| CGP | 49.70 | 264 (13) | 223 (8) | 3.390 | 3.159 | −0.46% |
| CGF | 50.31 | 276 (3) | 211 (13) | 2.92 | 3.29 | −0.76% |
| OSS | 51.91 | 209 (18) | 278 (12) | 2.78 | 4.75 | −3.95% |
| SCG | 52.73 | 157 (7) | 330 (8) | 1.71 | 4.51 | −5.61% |

[1,2] The number in the parenthesis is the number of grids that exceed the limits of the color bar. [3,4] The number of grids with a [3] positive/ [4] negative value of the difference of *RMSE*. [5] The change ratio is defined as (Improvement–Deterioration)/Total *RMSE*.

The comparison between RP ANN trained with synthetic database and hybrid database shows an improvement of 10.23% on the testing dataset overall. The performance was improved in 377 grids. The absolute value of the improvement is 6.51 m in total. However, 110 grids showed a higher *RMSE* value, contributing to a 1.41-m deterioration of performance in total.

The ANN trained with the hybrid database on the testing and RP had the lowest sum of *RMSE* of 44.83 m. The performance of CGB and CGP was the second and third best on the testing dataset, respectively. The SCG had the highest *RMSE* value of 52.73 m, which is 5.61% lower than the performance of the RP trained with the synthetic database.

### 3.2. Performance on the Real Events

These ANN were applied in four real events, and the average *RMSE* maps of real events were calculated. Similarly, the results from synthetic-database trained with RP were used as a benchmark. The map of the difference between the mean *RMSE* was produced, and all the statistics of the resulting maps are summarized in Table 2. Auxiliary maps are shown in figures in Figure A2.

**Table 2.** Statistics of the *RMSE* difference map for ANN on real events. Only the benchmark ANN, RP (SD), was trained with the synthetic database. Other ANN were trained with the hybrid database.

| Set Name | Sum of *RMSE* (m) | Improved Grids [1] | Deteriorated Grids [2] | Improvement [3] (m) | Deterioration [4] (m) | Change Ratio [5] |
|---|---|---|---|---|---|---|
| RP (SD) | 83.94 | - | - | - | - | - |
| RP | 69.68 | 229 (123) | 258 (51) | 23.79 | 9.54 | +16.99% |
| CGP | 77.66 | 198 (103) | 289 (77) | 19.57 | 13.28 | +7.49% |
| CGB | 81.12 | 184 (100) | 303 (90) | 18.99 | 16.16 | +3.36% |
| CGF | 83.83 | 209 (105) | 278 (81) | 20.01 | 19.90 | +0.13% |
| OSS | 92.25 | 164 (107) | 323 (187) | 19.94 | 28.25 | −9.89% |
| SCG | 99.46 | 162 (97) | 325 (213) | 18.43 | 33.95 | −18.49% |

[1,2] The number in the parenthesis is the number of grids that exceed the limits of the color bar. [3,4] The number of grids with a [3] positive/[4] negative value of the difference of *RMSE*. [5] The change ratio is defined as (Improvement–Deterioration)/Total *RMSE*.

The comparison between RP trained with the synthetic database and the hybrid database shows an improvement of 16.99% on the real events overall. The performance was improved in 229 grids. The absolute value of the improvement is 23.79 m in total. Nevertheless, there were 258 grids with lower performance, leading to 9.54-m deterioration in total.

For the performance of all the ANN trained by the hybrid database on the real events, the RP showed the best performance, with a sum of *RMSE* of 69.68 m. CGP and CGB had the second and third lowest sum of *RMSE* values on real events trained with the hybrid database. SCG had the highest *RMSE* value of 99.46 m, which is 18.49% lower than the performance of RP trained with the synthetic database.

## 4. Discussion

### 4.1. ANN on the Testing Dataset

Looking at the optimization of the training dataset during the testing phase, the hybrid database improved the RP performance by a satisfying level (Table 1). This is, in any case, expected, as the model is a data-driven model, and the training dataset was increased. Looking at the optimization of the training function (testing phase), it is clear that the RP training function led all the selected training functions (Table 2). As RP is the only algorithm that adopts the drawback elimination mechanism, the outstanding performance of it can likely be attributed to this unique feature. In this case, the downside of the partial derivatives' magnitude was eliminated during the backpropagation process, and as such, it did not affect the network training process (detailed explanation can be found in Section 2.3.2).

### 4.2. ANN on the Real Events

On the real events (Figure 5), it is noticeable that by using the hybrid database, the improvement of the performance on the real events was larger than on the testing phase (see Tables 1 and 2). Although the RP trained with the synthetic database already achieved relatively high accuracy on the testing phase, due to the simple nature of the synthetic database itself, the performance on the real events ranked below that of the testing phase. However, the hybrid database introduced rescaled historical events into the training, making the dataset more realistic. That is the reason why a further boost in the performance of the ANN was obtained on the real events.

For the optimization of the training function, on the real events, RP is the best choice, with some changes on the rankings of other functions (see Table 2). The CGF, the CGP, and the CGB had 290, 281, and 283 grids with improved performances, respectively; this means that they have more grids with lower *RMSE* compared to the RP. However, the sum of the *RMSE* value of improved grids is less than the sum of the *RMSE* value of deteriorated grids, which leads to their overall performance being lower than that of RP. This shows that although the RP is the best choice overall for the training function, regardless of the

dataset used (i.e., testing or real events), it is still possible that some grids show further improvements locally by using other training functions.

## 5. Conclusions

In this study, an ANN model for flood inundation forecast was improved by adopting an extended flood event database and an optimization of the training function. The study area of Kulmbach was divided into a 50 by 50 mesh covering 2500 grids. An ANN grid was constructed for each valid grid. The inputs for ANN grids were the hydrographs of three major rivers around the city, and the outputs were the water depth data of measure points in each grid.

Firstly, a hybrid database was adopted as the training dataset. The ANN were applied to four real events that occurred in Kulmbach to examine their performance. The results show that by using the hybrid database, the overall improvement of the performance is quite satisfying, showing more than 10% improvement in the accuracy and generality of the RP ANN. The result demonstrates that by adding the rescaled historical events to a synthetic event database, i.e., the hybrid database, the performance of the ANN improves.

Secondly, six common training functions were selected as candidates for performance testing: CGF, CGP, CGB, OSS, RP, and SCG, and they were tested to find out which is the best training function for ANN-based flood inundation forecasts. The results show that among all these six training functions, RP performed the best for 2D urban inundation forecasts, although it may not always be the best choice for every ANN grid.

In our case, the new training dataset reduced the model's *RMSE* by 10% and 16% for the testing dataset and real events, respectively; the selection of RP training function led to 15% lower *RMSE* for the testing dataset and 35% for the real events compared with the other five training functions. Hence, it is recommended that future studies should focus on improving the quality of the training dataset based on real events. This has a strong positive impact on the ANN performance. Calibrating and training the ANN with different functions can be laborious, especially when the ANN is large. In this case, RP can be the best choice both in terms of time and memory resources. Future work will look into other advanced machine learning methods, like ConvLSTM. This is an example of a method that is suitable to spatial-temporal prediction problems like flooding.

**Author Contributions:** Conceptualization, H.Z. and J.L.; methodology, H.Z.; software, H.Z.; validation, H.Z., J.L. and Q.L.; formal analysis, H.Z.; investigation, H.Z.; resources, J.L.; data curation, Q.L. and H.Z.; writing—original draft preparation, H.Z.; writing—review and editing, J.L.; visualization, H.Z.; supervision, J.L.; All authors have read and agreed to the published version of the manuscript.

**Funding:** This research received no external funding.

**Institutional Review Board Statement:** Not applicable.

**Informed Consent Statement:** Not applicable.

**Data Availability Statement:** The datasets generated for this study are not available due to access restrictions. The code in MATLAB is available upon request to the authors.

**Conflicts of Interest:** The authors declare no conflict of interest.

## Appendix A

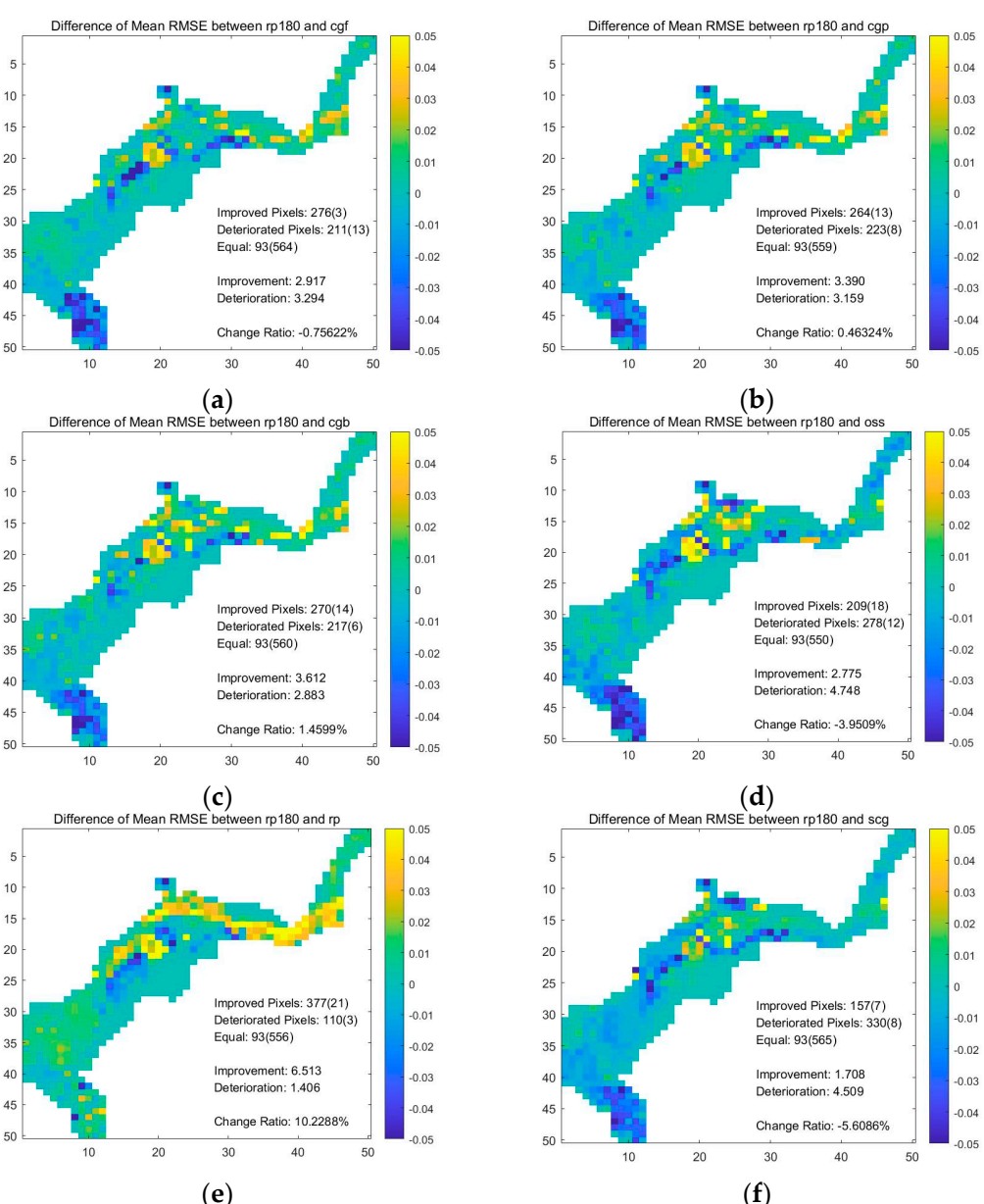

**Figure A1.** (**a**) Difference of mean *RMSE* on testing dataset between the set of RP (SD) and CGF ANN; (**b**) difference of mean *RMSE* on testing dataset between the set of RP (SD) and CGP ANN; (**c**) difference of mean *RMSE* on testing dataset between the set of RP (SD) and CGB ANN; (**d**) difference of mean *RMSE* on testing dataset between the set of RP (SD) and OSS ANN; (**e**) difference of mean *RMSE* on testing dataset between the set of RP (SD) and RP ANN; and (**f**) difference of mean *RMSE* on testing dataset between the set of RP (SD) and SCG ANN.

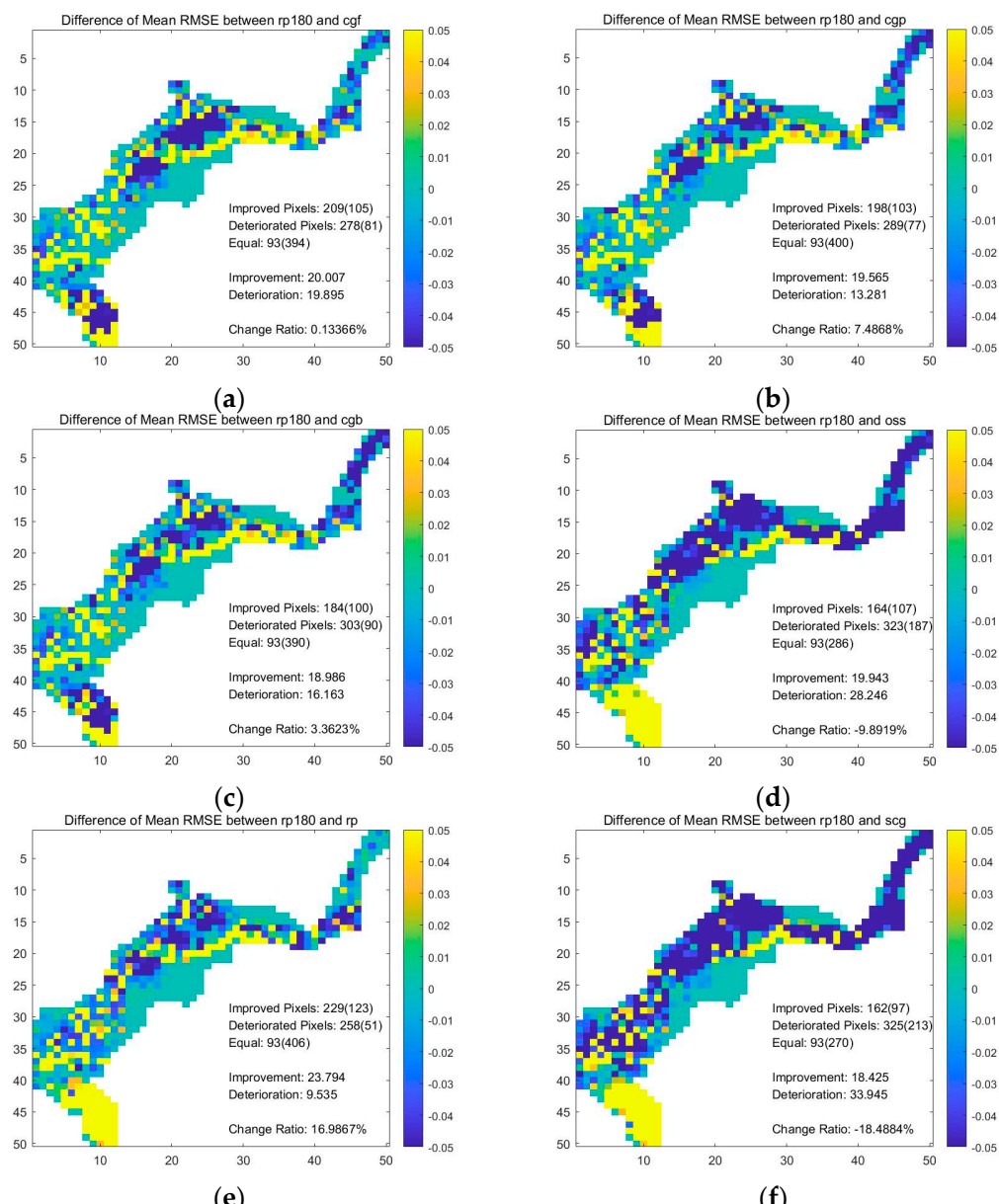

**Figure A2.** (**a**) Difference of mean *RMSE* on testing dataset between the set of RP (SD) and CGF ANN; (**b**) Difference of mean *RMSE* on testing dataset between the set of RP (SD) and CGP ANN; (**c**) Difference of mean *RMSE* on testing dataset between the set of RP (SD) and CGB ANN; (**d**) Difference of mean *RMSE* on testing dataset between the set of RP (SD) and OSS ANN; (**e**) Difference of mean *RMSE* on testing dataset between the set of RP (SD) and RP ANN; (**f**) Difference of mean *RMSE* on testing dataset between the set of RP (SD) and SCG ANN.

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
