# Peer review of "Optimization of Artificial Neural Network (ANN) for Maximum Flood Inundation Forecasts"

_water, doi:10.3390/w13162252_

Round 1

Reviewer 1 Report

Water-1301065 review:

Title: Optimization of Artificial Neural Network (ANN) for Maximum Flood Inundation Forecasts

Summary:

The authors presented a framework of optimizing Artificial Neural Network (ANN) by training the system based on synthetic flood events; extending the training datasets with the use of hybrid sets, and selecting the best trained function based on possible functions.

Reviewer comments:

I would like to preface my review by stating that my expertise is not of ANN, but more on urban flood modeling. I would like to prefer the Literature review and Methods be reviewed by another reviewer who is more familiar with the procedure, and hopefully get the suggestions from them. However, I can provide some additional suggestions with regards to the overall construction of the research paper and the general requirements for publication.

Major comments:

  1. Abstract

Provide a brief description of the methodology used

  1. Introduction

Provide short description of data-driven models and highlight the use of ANN for flood forecasting

  1. Methods

Show the map of the domain and the gauge stations.

  1. Discussion

Why are Figure A1 and A2 put in Appendix and not in Results?

  1. References

Follow MDPI format for citations: https://www.mdpi.com/authors/references.

Ex. Journal articles

Author 1; Author 2; Author 3; etc. Title of the article. Journal Abbreviation Year, Volume, Firstpage–Lastpage, doi:prefix/suffix.

Minor comments:

The whole paper needs to be proof read to fix minor formatting errors (Capitalization, inconsistent punctuation marks, referencing format).

Reviewer 2 Report

1 – The introduction section is a little bit lacking in proving the most recent studies in this area. The followings are good examples that can enrich the introduction section:

Tamiru, H., & Dinka, M. O. (2021). Application of ANN and HEC-RAS model for flood inundation mapping in lower Baro Akobo River Basin, Ethiopia. Journal of Hydrology: Regional Studies36, 100855.

Gavahi, K., Abbaszadeh, P., & Moradkhani, H. (2021). DeepYield: A Combined Convolutional Neural Network with Long Short-Term Memory for Crop Yield Forecasting. Expert Systems with Applications, 115511.

Zhou, Y., Wu, W., Nathan, R., & Wang, Q. J. (2021). A rapid flood inundation modelling framework using deep learning with spatial reduction and reconstruction. Environmental Modelling & Software, 105112.

2 – The authors have chosen the size of 50 for their grid size. What is the effect of a higher or lower grid size? Why not choosing 25?

3 – “After that, the appropriate number of neurons and hidden layers are 107 determined by varying the number of hidden layers from 2 to 12, and the neurons number 108 in each layer from 10 to 60” what is the rational behind choosing these numbers. I suggest that authors elaborate more on the reasons for choosing these values.

4 – How did the author selected the nearby river for each pixel? What distance is considered as near?

5 – I suggest that authors consider ConvLSTM for their future works since the method is most suitable for flood inundation mapping applications.

Round 2

Reviewer 2 Report

The authors have fully addressed my comments and I suggest the manuscript publication in Water.